# Multimodal Graph-LLM: Leveraging Graph-Enhanced LLMs for Multimodal Healthcare Predictions

## Abstract

Multimodal healthcare research is crucial for improving clinical decision-making by integrating diverse data types, such as clinical notes, lab results, and imaging. Large Language Models (LLMs) are widely recognized for their exceptional text-based reasoning capabilities, making them effective in processing complex clinical narratives. However, they struggle to incorporate multimodal data, limiting their broader applicability in healthcare analysis. In this work, we propose `MG-LLM` (Multimodal Graph-LLM), a novel framework that leverages the strengths of LLMs while enhancing them with multimodal alignment and data integration through Graph Neural Networks (GNNs). GNNs propagate information across similar patients, model temporal relationships between visits, and align information from different modalities, creating enriched multimodal context vectors. These context vectors are then injected into the intermediate layers of the LLM, allowing it to harness both textual reasoning and multimodal data for more accurate predictions. We evaluate `MG-LLM` on the MIMIC-IV and MIMIC-CXR datasets, demonstrating significant improvements in clinical prediction tasks compared to baseline models. Our results showcase the potential of combining the text reasoning power of LLMs with GNN-driven multimodal alignment for robust, comprehensive healthcare analysis.

## 1 Introduction

Electronic Health Records (EHR) data provide a multimodal representation of a patient's health, encompassing medical images, unstructured data such as clinical notes, codified data such as ICD-9/10/11 codes, and structured data such as labs or vitals as shown in Figure 1 (Hoerbst and Ammenwerth, 2010). The integration of these diverse data types offers great promises in enhancing clinical predictions in terms of more accurate and holistic patient assessments (Huang et al., 2020; Kim, 2022; Zhang et al., 2019; Xu et al., 2021; Yang et al., 2021; Moshawrab et al., 2023). However, the complexity and heterogeneity of EHR data present unique analytical challenges, particularly in effective representation and integration of patient information across heterogeneous data modalities.

Large language models (LLMs) have the potential to make a transformative impact on healthcare due to their exceptional capabilities in processing and reasoning over text data. These models excel in tasks such as summarizing clinical notes, extracting medical conditions, and generating clinical recommendations(Nazi and Peng, 2024a). Despite these advancements, LLMs face limitations when applied to complex, heterogeneous multimodal healthcare data. Two major challenges arise when considering the use of LLMs for clinical predictions.

**Limitations of LLMs with Multimodal EHR Data.** LLMs have demonstrated remarkable capabilities in processing textual data, but they face significant challenges when dealing with multimodal EHR data (Nazi and Peng, 2024a; Kim, 2022; Zhou et al., 2024). These challenges arise from the inherent design of LLMs, which are primarily optimized for handling sequential text data. This focus on text restricts their ability to seamlessly incorporate non-textual data such

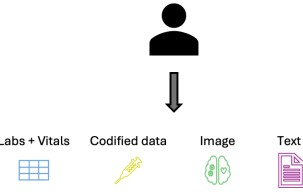

Labs + Vitals   Codified data   Image   Text

Figure 1: Multimodality in EHR.

as medical images, lab results, and structured codes like ICD codes, resulting in incomplete or fragmented patient representations and underutilization of the rich, multimodal information available in EHRs (Nazi and Peng, 2024b; Yildirim et al., 2024). Although a few multimodal LLMs have been introduced (He et al., 2024; Song et al., 2023), they often require additional frameworks or specialized architectures to align and contextualize information from other modalities, such as medical images or labs and vitals. This approach does not fully leverage the integrated reasoning and language capabilities inherent to LLMs (Mumtaz et al., 2024) and has not yet been actively explored in EHR data (Huang et al., 2023).

Another limitation is the LLMs' inability to perform context-dependent interpretation of medical data. For instance, they may struggle to determine the clinical significance of a slight elevation in a lab value for different patients with varying health contexts. In real-world diagnostic tasks, these limitations lead to a performance gap between LLMs and human physicians. LLMs often underperform in synthesizing diagnostic information, following treatment guidelines, and integrating diverse data sources that are crucial for accurate clinical predictions (Hager et al., 2024). Despite these challenges, the potential of LLMs in healthcare remains significant, underscoring the need for continued development to enhance their ability to process diverse data types and maintain the contextual understanding necessary for effective clinical decision-making.

**Limitations of LLMs in Leveraging Patient Similarity.** Learning from the experiences of similar patients is a critical aspect of clinical decision-making, as healthcare professionals often draw upon patterns observed in patients with comparable conditions to inform diagnoses and treatment plans (Zhang et al., 2022a). For example, when assessing a patient's symptoms, physicians may consider how patients with similar medical histories responded to certain treatments or exhibited particular disease progressions.

In contrast, LLMs process each patient case as an isolated input during inference. They rely on the generalized knowledge encoded in their training data to generate responses, without explicitly referencing individual patient cases or directly accessing a repository of similar patients' information. This is because LLMs do not maintain a dynamic, context-aware memory of specific cases but instead generate responses based on probabilistic patterns learned from a broad dataset (Bommasani et al., 2022). While this approach allows LLMs to generalize across various scenarios and generate plausible responses, it may miss critical, nuanced similarities that are often essential in healthcare settings. Moreover, healthcare decision-making often involves evaluating complex, multifaceted patterns across time, considering not only the current symptoms but also how these symptoms have evolved in relation to past medical events (Ageno et al., 2023). The temporal nature of clinical data, along with the need to consider information from similar patients, requires a more sophisticated approach than the static, generalized knowledge offered by current LLMs.

To address these limitations, we propose `MG-LLM` ((Multimodal Graph-LLM), which leverages the language-processing and reasoning capabilities of LLMs while seamlessly integrating multimodal patient data. Our approach aims to enhance patient representation, improve multimodal data integration, and enable better contextual reasoning. The key contributions of our work are as follows.

- ⋆ **Dynamic Patient Similarity Integration:** A graph-based approach that explicitly models patient similarities and temporal relationships, enabling LLMs to make more informed, context-aware decisions.

- ⋆ **Multimodal Data Integration:** Align multimodal data into a unified representation that can be seamlessly injested and utilized by LLMs.

- ⋆ **Improved Contextual Reasoning:** Preserve the LLM's strong text-based reasoning abilities while augmenting it with multimodal information.

## 2 RELATED WORKS

### 2.1 MULTIMODAL APPROACH

Existing multimodal LLM approaches typically involve the use of modality-specific encoders that transform different data types, such as images, time-series data, and structured clinical data, into a shared representation space compatible with LLMs. Some methods use encoders to transform

high-dimensional health data, such as spirograms and lab values, into the LLM's token embedding space, allowing the model to process these inputs along with text data (Belyaeva et al., 2023). Similarly, other approaches integrate clinical notes with structured data by encoding lab results into text-like embeddings to facilitate integration with LLMs (Ding et al., 2024). However, this approach may limit the LLM's ability to fully leverage its strengths in text-based reasoning. Other methods directly convert clinical data into textual descriptions (e.g., representing lab values as text), but this limits their ability to handle non-textual modalities like medical images effectively (Rezk et al., 2024). Despite their innovative designs, these approaches face common issues. They often treat each modality separately before concatenating the resulting embeddings, which can lead to fragmented representations and reduce the LLM's ability to leverage its full language-based reasoning capabilities. These challenges highlight the need for more cohesive frameworks that can more effectively integrate and reason over diverse healthcare modalities.

Various non-LLM multimodal frameworks have also been developed to tackle the challenges of integrating diverse medical data types. Graph-based approaches have been particularly effective in modeling the complex relationships within EHR data, enabling the flexible integration of different data modalities (Gao et al., 2020; Zhang et al., 2022a). Some methods focus on addressing the common issue of missing data in medical records by enhancing the robustness of multimodal analysis through sophisticated data imputation techniques (Zhang et al., 2022a). Temporal aspects of patient data have also been considered, with several models designed to handle sequential multimodal information, thereby providing a more comprehensive view of patient health over time (Wu et al., 2024). Additionally, other approaches have concentrated on integrating specific modalities prevalent in healthcare, such as combining deep learning models for medical image analysis with architectures designed to handle structured data, thereby offering a holistic perspective on patient health (Zhang et al., 2020; Xu et al., 2021). These advancements reflect the growing emphasis on comprehensive data integration to support more informed and accurate clinical decision-making. While these models have made significant strides in handling different types of healthcare data, they often lack the deep language understanding and contextual reasoning capabilities that LLMs possess.

## 2.2 GRAPHS AND LLMs

Recent work has made notable progress in combining LLMs with graph-based methods to leverage the structured knowledge encoded in graphs alongside the language understanding capabilities of LLMs. Techniques like Graph Neural Prompting and the LLaGA framework integrate LLMs with graph neural networks (GNNs) to encode and align graph data, significantly enhancing the LLMs' performance on tasks such as question answering and knowledge retrieval (Tian et al., 2023; Chen et al., 2024). However, these approaches mainly address single-modality graphs or text data, with limited exploration of multimodal graphs that incorporate diverse data types like images, clinical notes, and structured clinical data. While models like LLM4GraphGen (Yao et al., 2024) and Talk Like a Graph(Fatemi et al., 2023) have advanced graph generation and encoding for LLMs, they largely focus on single-modality settings, highlighting a need for research on integrating LLMs with multimodal graphs to fully utilize the rich, heterogeneous data available in fields such as healthcare.

## 3 OUR APPROACH: `MG-LLM`

In this section, we introduce `MG-LLM`, a framework specifically designed to leverage multimodal data with LLMs. The process begins by constructing graphs for each modality, such as codified data, lab results, and imaging data. We then apply GNNs to propagate information across these graphs. Next, we align the updated embeddings across modalities to generate a comprehensive context vector for each patient. This context vector is injected into the intermediate layers of the LLM, which is simultaneously prompted with clinical notes to predict clinical outcome of interest, such as 1-year mortality risk. An illustration of `MG-LLM` is provided in Figure 2.

### 3.1 INFORMATION PROPAGATION

We generate embeddings for each modality using pre-trained encoders. For codified data and clinical text, embeddings are obtained using BioClinicalBERT, a model specifically trained on clinical data to capture the nuances of medical language (Alsentzer et al., 2019). For the imaging data,

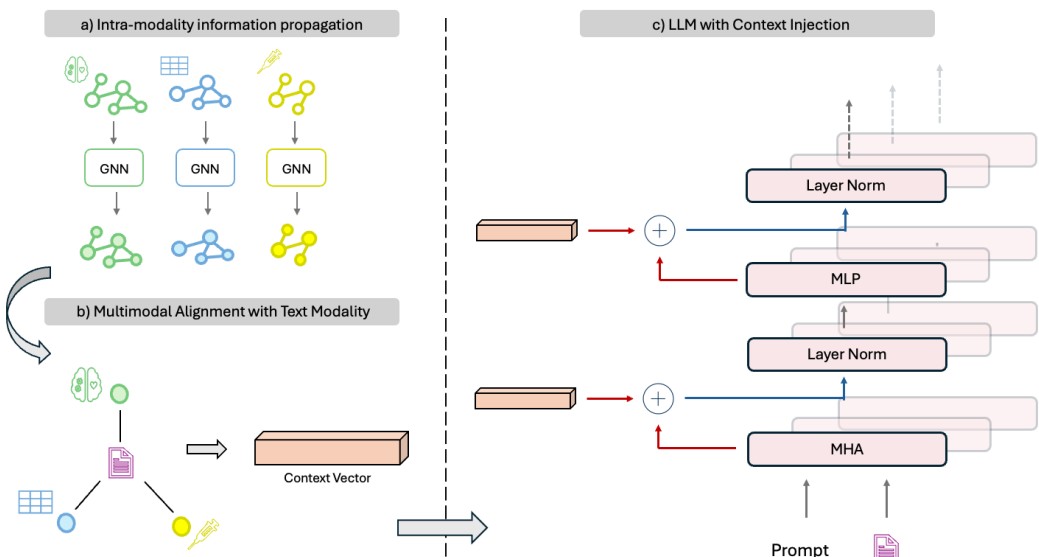

Figure 2: Illustration of `MG-LLM`. (a) For each modality, we construct a graph with similarity edges and temporal edges. Then, we perform information propagation via GNNs to get the updated node embeddings. (b) We perform multimodal alignment using text as the central modality to generate a patient-level context vector. (c) We inject the context vector into the intermediate layers of the LLM and prompt the LLM using the clinical text. Thus, we preserve the text-based reasoning capabilities while augmenting the LLM with multimodal context to increase its performance.

a ResNet50 model pre-trained on ImageNet is utilized to extract image features and generate the corresponding embeddings (He et al., 2015). Other pre-trained encoders can also be used.

Once the embeddings are obtained for each data point, we construct a separate graph for each modality. Each graph represents a distinct data modality, with nodes corresponding to data points (i.e. patients)and edges representing relationships between these nodes. We introduce two types of edges: temporal edges and similarity edges. Temporal edges connect nodes that correspond to the same patient across different time points, thereby capturing the longitudinal nature of patient care. This allows the model to track changes in a patient's health over time, ensuring that any temporal trends or patterns are incorporated into the model's predictions.

In addition to temporal edges, similarity edges are established by calculating the cosine similarity between the embeddings of different data points. These similarity edges link nodes based on shared characteristics, even if the nodes represent different patients. This enables the graph structure to propagate information not only within the same patient (across visits) but also across different patients who exhibit similar patterns. By doing so, the model can learn from the experiences of similar patients, improving its predictive power.

The combination of temporal and similarity edges in the graph structure enables efficient information propagation through a GNN. The GNN allows information to flow through these connections, effectively transferring information both within a single patient over time and across different patients. As a result, the model can capture both the temporal dynamics of a patient's health and broader population-level patterns, providing a more comprehensive representation of the data and improving the accuracy of downstream predictions, such as mortality risk or disease progression.

## 3.2 MULTIMODAL ALIGNMENT

To align the embeddings across different modalities, we draw inspiration from the ImageBind framework (Girdhar et al., 2023). ImageBind is a state-of-the-art framework that creates a shared embedding space for multiple modalities—such as images, audio, and text—by using images as the central modality for alignment. This approach facilitates cross-modal retrieval and understanding by learning a shared representation that captures relationships between these diverse data types. ImageBind's

strength lies in its ability to bridge these modalities into a common space, enabling effective cross-modal interactions.

We adapt this idea to the healthcare domain by using clinical text as the primary modality for alignment instead of images. The choice of "text modality" as central modality is motivated by its ability to provide a comprehensive narrative of a patient's conditions, capturing nuances that are often not fully reflected in "non-text modalities" such as codified data (e.g., diagnosis codes) and medical images. Furthermore, utilizing clinical text allows us to leverage the powerful language processing capabilities of LLMs, which excel in understanding and generating insights from complex text data.

To achieve effective multimodal alignment, we first obtain updated embeddings from the modality-specific graphs. Through contrastive learning, we map the non-text modality embeddings (codified data, labs and vitals, medical images) and the text modality embeddings into a shared projection space. This alignment process ensures that embeddings from different modalities are represented in a common latent space, enabling more effective cross-modal interactions and information sharing. The shared projection space is optimized using an InfoNCE loss (Parulekar et al., 2023). Given text data $T_i$ and another modality data $M_i$, we compute the embeddings $t_i$ and $m_i$ and the loss as:

$$L_{T,M} = -\log \frac{\exp\left(\frac{t_i^\tau m_i}{\tau}\right)}{\exp\left(\frac{t_i^\tau m_i}{\tau}\right) + \sum_{j \neq i} \exp\left(\frac{t_j^\tau m_j}{\tau}\right)} \tag{1}$$

This loss ensures that embeddings from matching modalities (text and non-text data for the same patient) are drawn closer together in the shared space, while embeddings from different patients or unrelated modalities are more separated. The use of this approach enhances the interaction between different types of data, ensuring that complementary insights from various modalities are aligned effectively. Once the different modalities are mapped into the common embedding space, we use a linear layer to aggregate these individual embeddings into a single patient-level context vector. This context vector encapsulates the comprehensive, holistic information from all the modalities (clinical text, codified data, lab results, and images) into a unified representation of the patient's medical profile.

## 3.3 INCORPORATING MULTIMODAL CONTEXT IN LLMs

Once we generate a multimodal context vector that encapsulates information from various data sources, the next step is to integrate this context into the LLM in a way that enriches its predictions without altering its fundamental architecture. This is achieved by injecting the multimodal context vector into the LLM's residual streams, which would allow the model to incorporate external information at various layers while preserving its inherent structure and reasoning capabilities (Li et al., 2024). The rationale behind this approach is to maintain the LLM's core strengths in natural language processing while enhancing its ability to process non-textual data, such as medical images or lab results, which are crucial for clinical decision-making. By incorporating this multimodal information in the form of a context vector, we enable the LLM to use structured and unstructured medical data to make more informed and contextually rich predictions, particularly in scenarios such as predicting a patient's mortality risk.

The integration of the context vector into the LLM occurs at multiple points within its internal architecture. Let $\mathbf{r}_t^l$ represent the residual stream at layer $l$ and token position $t$. Normally, the LLM adds the output activations from the Multi-Head Attention (MHA) and Multi-Layer Perceptron (MLP) modules to the residual streams at each layer. However, in our approach, we enhance this mechanism by introducing a weighted linear combination of the patient's context vector with the MHA and MLP activations. This step is expressed as:

$$r_t^l = r_t^{l-1} + \lambda_{mha} \cdot v + \beta_{mha} \cdot a_t^l + \lambda_{mlp} \cdot v + \beta_{mlp} \cdot m_t^l \tag{2}$$

where $\lambda_{mha}$ and $\lambda_{mlp}$ are scaling factors for the context vector integration into the MHA and MLP layers. $\beta_{mha}$ and $\beta_{mlp}$ are scaling factors for the original MHA and MLP outputs. We apply this process to all residual streams using the forward hook. By injecting the context vector into the residual streams in this manner, the LLM can leverage external context effectively and efficiently while preserving its inherent architecture and text-based reasoning capabilities. This approach allows us to integrate non-text modalities into the model while only providing text as input to the LLM, ensuring that we retain the LLM's core strength in text-based reasoning.

For clinical prediction, our approach is as follows using mortality prediction as an example. When prompting the LLM, we task it with estimating the patient's probability of experiencing mortality within a year. The model is provided with the relevant clinical text as input, while the multimodal context vector is injected into its layers to support the decision-making process. This framework ensures that the LLM has access to a wealth of external context beyond the text alone, enabling it to provide more accurate and nuanced predictions. By integrating text and non-text modalities seamlessly within the LLM, we retain the model's strengths in natural language processing and text reasoning while enhancing its capability to incorporate complex, multimodal medical data and hence more comprehensive patient medical history.

## 4 EXPERIMENTS

We conducted real data experiments to assess the performance of `MG-LLM` in comparison with current state of the art models.

### 4.1 EXPERIMENTAL SETTING

**MIMIC-IV Data.** MIMIC-IV (Medical Information Mart for Intensive Care IV) is a large, publicly available dataset containing de-identified health-related data associated with patients who stayed in critical care units at the Beth Israel Deaconess Medical Center (Johnson et al., 2023). This dataset includes structured data such as demographics, vital signs, laboratory measurements, medications, and procedures and also unstructured data in the form of clinical notes. MIMIC-CXR is an open-access database of de-identified chest X-ray images paired with radiology reports that can be linked to the MIMIC-IV dataset (Johnson et al., 2019). For the purposes of this experiment, we extract labs and vital values, codes for prescriptions and diagnoses, clinical text, and imaging data. Our prediction task is a binary classification of one-year-mortality for the patient. For patient $A$ with visits $v_1, v_2, ..., v_n$, we will use information from all of the visits to predict risk of mortality within one year timeframe after $v_n$. More details about dataset processing is in Appendix.

**Baseline Models.** For existing state-of-the-art models, we include multimodal LLM baselines and other state-of-the-art frameworks designed for clinical prediction tasks. Multimodal LLM frameworks, such as HeLM, LLMM, and GPT-4-based methods, explore various approaches to integrating multimodal data with LLMs (Belyaeva et al., 2023; Ding et al., 2024; Rezk et al., 2024). HeLM encodes non-text modalities into the same token embedding space using modality-specific encoders and then inputs them to the LLM (Belyaeva et al., 2023). LLMM combines laboratory test data with text embeddings through an attention mechanism (Ding et al., 2024). Rezk et al. transform clinical data into text format for input into the LLM (Rezk et al., 2024). For non-LLM based frameworks, the HAIM framework leverages the integration of multiple data modalities, such as clinical notes, structured data, and medical images, to improve clinical predictions through a unified model architecture (Soenksen et al., 2022). M3Care focuses on learning with incomplete modalities in multimodal healthcare data by employing neural networks and patient similarity measures (Zhang et al., 2022a). MUSE incorporates flexible bipartite graphs and contrastive learning loss to generate multimodal patient representations (Wu et al., 2024). mmFormer introduces a transformer-based approach to multimodal fusion, using attention mechanisms to align and integrate information from multiple modalities, thereby capturing complex relationships in the data (Zhang et al., 2022b). Collectively, these frameworks provide comprehensive baselines for evaluating the performance of our proposed method, given their strong track records in handling diverse and complex multimodal healthcare data.

**Implementation Details.** The dataset was split into 70% training and 30% testing. For `MG-LLM`, we trained the entire framework end to end. During training, we used a combination of two losses: a weighted binary cross-entropy loss and the alignment loss, which was scaled by 0.01 to appropriately adjust its contribution within the overall task. To handle any non-valid outputs generated by the LLM, we imposed penalties on outputs that fall outside the expected prediction range, ensuring the model remains within valid boundaries. `MG-LLM` uses Llama 3-8B as its LLM backbone. For the baseline experiments, we used the best hyperparameter setting provided in the original papers. If the hyperparameters were not available, we tuned the learning rate in 1e-3, 1e-4, 1e-5 and batch size in 8, 16, 64. All experiments were conducted using NVIDIA A100 GPUs. Each experiment was run three times with different seeds to ensure reproducibility, and we averaged the results.

|  | LLM-based | Lab+Vital | Code | Image | Text | Acc | F1 |
|---|---|---|---|---|---|---|---|
| HAIM | ✗ | ✓ | ✓ | ✓ | ✓ | 71.15±0.00 | 58.45±0.00 |
| M3Care | ✗ | ✓ | ✓ | ✓ | ✓ | 77.72±2.21 | 55.53±0.90 |
| MUSE | ✗ | ✓ | ✓ | ✗ | ✓ | 77.40±1.12 | 51.25±1.87 |
| mmFormer | ✗ | ✓ | ✓ | ✓ | ✓ | 76.52±0.79 | 60.93±2.41 |
| HeLM | ✓ | ✓ | ✗ | ✗ | ✓ | 74.78±1.16 | 51.67±0.15 |
| LLMM | ✓ | ✓ | ✗ | ✗ | ✓ | 75.75±0.07 | 48.12±0.05 |
| (Rezk et al., 2024) | ✓ | ✓ | ✓ | ✗ | ✓ | 75.10±0.09 | 65.42±1.07 |
| Llama 3-8b | ✓ | ✗ | ✗ | ✗ | ✓ | 69.4±0.00 | 50.6±0.00 |
| **MG-LLM** | ✓ | ✓ | ✓ | ✓ | ✓ | **78.8**±0.97 | **64.6**±1.21 |

Table 1: Comparison of models in terms of key model characteristics and predication performance measured by accuracy and F1 score.

## 4.2 PRIMARY RESULTS

In Table 1, we provide a comprehensive comparison of `MG-LLM` against the baseline models of interest including both LLM-based and non-LLM-based approaches, in terms of their key characteristics and predication performance. Table 1 highlights the impact of incorporating different modalities—lab results, codes, images, and text—on the performance of each model, as measured by accuracy and F1 score. Several key observations can be made from our results.

`MG-LLM` achieves the highest combined performance across both accuracy and F1 score, with an accuracy of 78.8% and an F1 score of 64.6. While one baseline model, Llama 3-8b, has a slightly higher F1 score (65.42), it achieves a significantly lower accuracy (75.10%), indicating a trade-off between these two metrics. In contrast, `MG-LLM` strikes a more effective balance between accuracy and F1 score, demonstrating superior robustness in handling complex multimodal inputs such as lab, code, image, and text data.

Interestingly, models such as HeLM and LLMM, which are also LLM-based, show lower overall performance. For instance, HeLM achieves an accuracy of 74.78% but has a relatively lower F1 score of 51.67, which suggests that even though it handles multimodal inputs, the lack of integration of all relevant modalities (e.g., image) hampers its overall classification ability. Similarly, LLMM, despite integrating multiple modalities, achieves a lower F1 score of 48.12, indicating that modality integration alone is not sufficient for superior performance without an effective multimodal strategy.

Moreover, non-LLM models such as M3Care and mmFormer show competitive performance but still fall short in terms of overall accuracy and F1 when compared to `MG-LLM`. M3Care achieves 77.72% accuracy but lags behind with a F1 score of 55.53, indicating that while non-LLM models can perform well, they do not leverage the full reasoning potential of LLMs when combined with multimodal data.

The ability of `MG-LLM` to incorporate all relevant modalities and leverage the reasoning capabilities of an LLM results in more informed predictions, leading to its superior performance. By effectively integrating lab results, codes, images, and text, `MG-LLM` maximizes the complementary information provided by each modality. This allows it to make better overall predictions, particularly when dealing with complex medical datasets, where multimodal input is crucial for accurate classification.

In summary, the performance gain by `MG-LLM` can be attributed to its ability to incorporate all relevant multimodal data, effectively balancing accuracy and F1 score. This indicates that the multimodal context vector injection, combined with the reasoning power of an LLM, plays a crucial role in delivering superior classification performance compared to both LLM-based and non-LLM-based models in this comparison.

## 4.3 ABLATION STUDIES

We also examined the impact of removing some data modalities on the performance of `MG-LLM` for which the results are reported in Table 2. This analysis shows how each data modality contributes to the inference, highlighting the importance of multimodal frameworks when working with healthcare data. From Table 2, the first key observation is that performance improves as the number of modalities increases. The highest performance, in terms of both accuracy and F1 score, is achieved when all four modalities—lab, code, image,

| Lab+Vital | Code | Image | Text | Acc | F1 |
|:---:|:---:|:---:|:---:|:---:|:---:|
| ✗ | ✓ | ✗ | ✓ | 75.88 | 56.41 |
| ✓ | ✗ | ✗ | ✓ | 76.12 | 57.72 |
| ✗ | ✗ | ✓ | ✓ | 74.85 | 53.18 |
| ✗ | ✓ | ✓ | ✓ | 77.88 | 58.06 |
| ✓ | ✗ | ✓ | ✓ | 76.44 | 59.41 |
| ✓ | ✓ | ✗ | ✓ | 77.37 | 60.02 |
| ✓ | ✓ | ✓ | ✓ | 78.80 | 64.60 |

Table 2: Ablation study with different data modality combinations

and text—are used together. This suggests that each modality provides complementary information, and combining them results in the most robust predictions. It is also important to note the critical role of lab and code data. Performance drops significantly when these two modalities are excluded. The lowest performance is observed when only image and text data are used, indicating that lab and code data contribute essential structured information that is crucial for accurate decision-making. In particular, the combination of text and image alone (row 3) results in the poorest performance, with both accuracy and F1 scores reaching their lowest values. This can be attributed to the complexity of interpreting medical images and the limitations of relying solely on text data, which may not provide sufficient context without the support of structured data such as lab results and diagnosis codes. Overall, these results clearly demonstrate the importance of a multimodal approach. Lab and code data provide vital structured information, while text and image data add valuable but complementary insights. To achieve the best performance when working with complex medical datasets like MIMIC, it is essential to leverage all available data modalities.

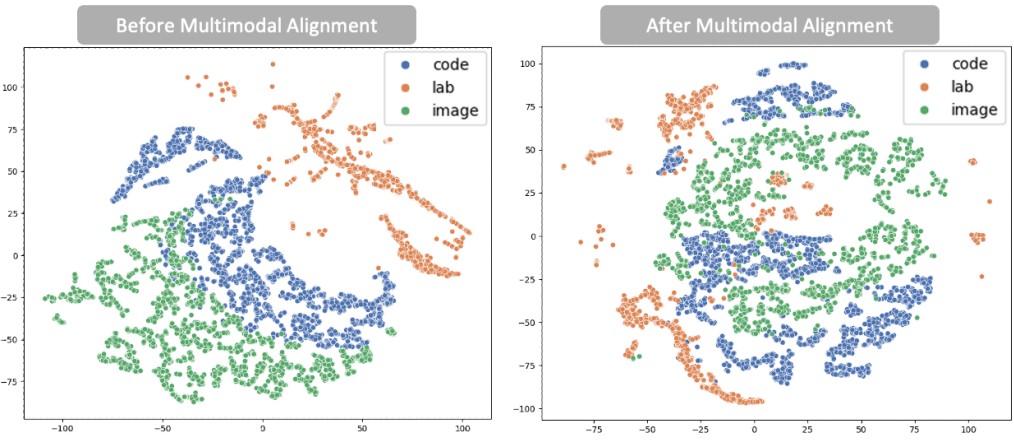

Figure 3: tSNE plot of modality embeddings before and after multimodal alignment

## 4.4 EFFECTIVENESS OF MULTIMODAL ALIGNMENT

In Figure 3, the plot on the left shows the distribution of embeddings for three different modalities—code (blue), lab (orange), and image (green)—prior to multimodal alignment. The clear separation between the clusters for each modality indicates that the embeddings occupy distinct regions of the feature space. As a result, each modality functions independently, and there is limited, if any, cross-modal information sharing. The distinct boundaries between the modalities highlight the challenge of integrating multimodal data without a shared representation, which limits the ability of the model to fully exploit the complementary nature of the data.

In contrast, the plot on the right demonstrates the effectiveness of the multimodal alignment process. After alignment, we observe that the embeddings from the different modalities are now more inter-

mixed in the feature space, with the code, lab, and image embeddings much closer together. This intermixing is a visual representation of the successful alignment of the modalities into a shared latent space. The fact that the clusters have blended suggests that the model has learned to map each modality to a common representation where cross-modal relationships can be identified and utilized.

The improved proximity between the modalities in the aligned space signifies that the model can now effectively integrate information from code, lab, and image data, enhancing its ability to make informed predictions based on a more comprehensive view of the patient's health. This blending of the embeddings demonstrates that the contrastive learning approach used in the alignment process has succeeded in aligning the modalities. Consequently, the multimodal context vector generated for each patient contains information from all modalities in a coherent, shared space, which can then be injected into the LLM to improve its performance. Overall, this result highlights the value of multimodal alignment, as it transforms previously disparate modalities into a unified, integrated feature space where cross-modal interactions are not only possible but optimized. This alignment ensures that the LLM can leverage the strengths of each modality, ultimately enhancing its predictive accuracy in complex tasks such as those in clinical decision-making.

| Output | Explanations |
|---|---|
| Positive Mortality Prediction | The patient has a history of coronary artery disease (CAD), multiple myocardial infarctions (MIs), including a recent NSTEMI, and congestive heart failure (CHF) with a severely reduced ejection fraction (EF) of 15% as noted in his most recent transthoracic echocardiogram (TTE). He is currently on a heparin bridge to Coumadin with a target INR of 2.5, as well as several cardiac medications, including a statin, an ACE-inhibitor, a beta-blocker, and a diuretic. Although his intake and output (I&O) balance is stable, his overall clinical status, including fluid and sodium restrictions, reflects the severity of his condition. |
| Negative Mortality Prediction | This patient is at a relatively low risk of mortality. She is a 45-year-old woman with a history of breast cancer, currently undergoing treatment, as well as well-managed hypertension and GERD. Although she is a candidate for a hysterectomy and oophorectomy due to a diagnosis of ovarian cancer, her overall health is stable, and there are no immediate signs of life-threatening complications. Her vital signs are stable, and her comorbidities, such as hypertension and GERD, are controlled with medication. |

Figure 4: Examples of `MG-LLM` generating explanations for both positive and negative mortality risk predictions.

## 4.5 EXPLAINABILITY

A key advantage of LLMs over other multimodal frameworks is their ability to not only generate predictions but also provide natural language explanations for the decisions that they make. This capability is particularly appealing and valuable in clinical settings, where transparency and understanding the reasoning behind predictions are essential for building trust in health professionals, patients and other stakeholders. As shown in Figure 4, `MG-LLM` generates detailed, context-specific explanations for both positive and negative mortality predictions. For instance, when predicting high mortality risk, the model provides a comprehensive explanation citing the patient's history of coronary artery disease, multiple myocardial infarctions, and other clinical details that justify the prediction. In contrast, for a negative mortality prediction, the model explains how the patient's stable condition, well-managed comorbidities, and absence of immediate life-threatening complications contribute to the outcome. This level of detail in the explanations enhances interpretability and allows clinicians to better understand and verify the model's reasoning. In comparison, non-LLM baseline models lack this ability to provide transparent, context-specific reasoning, underscoring the distinct advantage of using LLMs in high-stake decision making in areas such as healthcare.

## 5 CONCLUSION

Multimodal healthcare data holds a wealth of valuable information that, which, when integrated effectively, can enhance clinical decision-making. However, existing approaches that do not utilize LLMs fail to capitalize on their powerful reasoning capabilities, while current LLM-powered approaches often overlook key modalities like medical images. Our work introduces `MG-LLM`, a flexible framework that bridges these gaps by propagating information across patients and injecting multimodal context into LLMs. By incorporating clinical notes, lab results, codified data, and images into LLMs, `MG-LLM` retains the model's strengths in text-based reasoning while enhancing it by utilizaing other data modalities to create a more comprehensive context. The demonstrated improvements over other methods in our experiments show `MG-LLM`'s effectiveness in leveraging multimodal data. Future research may include more comprehensive assessment of explanations generated by `MG-LLM` and in additional EHRs datasets such as those from primary care settings.

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

# A APPENDIX

## A.1 DATASET PROCESSING

We first process the MIMIC-IV dataset using the PyHealth package. The PyHealth package automatically extracts patient demographic information, hospital admission information, and ICD9 codes such as prescriptions, drugs, and diagnoses. To extract clinical text, we reference the discharge.csv file. Laboratory results were referenced from labevents.csv file in the MIMIC-IV dataset. To link the MIMIC-IV and MIMIC-CXR datasets, we use the unique identifiers studyid, hadmid, and subjectid available in both datasets. For one-year mortality labels, we predict whether the patient will experience mortality within one year after their most recent visit recorded in the dataset. Due to computational constraints as we need to prompt the LLM for each datapoint, we use a downsampled dataset. The rate of positive examples is 0.23.

