# OpenReview forum: "Multimodal Graph-LLM: Leveraging Graph-Enhanced LLMs for Multimodal Healthcare Predictions"
_ICLR.cc/2025/Conference — ICLR 2025 Conference Withdrawn Submission_

### Official Review · Reviewer_sz7M · 2024-10-23

**Soundness:** 2
**Presentation:** 2
**Contribution:** 2
**Rating:** 3
**Confidence:** 4

**Summary:**

The paper presents a framework that integrates the reasoning capabilities of LLMs with GNNs on the assumption that GNNs have the multimodal data processing power.  With this design, the authors hope the work will enhance clinical predictions by incorporating diverse healthcare data types such as clinical notes, lab results, and medical images.
In addition, the proposed method also leverages GNNs to model patient similarities and temporal relationships while aligning multimodal information into a unified context vector that enriches the LLM. The framework is evaluated on MIMIC-IV and MIMIC-CXR datasets, showing decent performance over baseline models, but worse than (Resz at al 2024) from results in Table 1.

**Strengths:**

- The intention on multimodality is meaningful as health data is inherently multimodal and no single modality can fully capture comprehensive health and patient information.
- Using GNNs for integrating multimodal data is effective and makes sense.
- the dynamic patient similarity modeling using GNNs is a good approach.

**Weaknesses:**

- Very limited dataset, use case, and evaluation tasks, which limits the generalizability of the model to other datasets and healthcare environments.. MIMIC data was collected within critical care scenario with very limited use cases. Even though, previous research often evaluated their works across diverse set of tasks including readmission prediction, next dat discharge prediction, treatment recommendation, medical coding, and so on. However in this work, the authors only consider mortality prediction as the evaluation task. This is not sufficient to demonstrate the utility of the proposed approach.
Note that from Table 1, even with multimodal data, the proposed approach is not the best although authors make their proposed model bold font to indicate best performance.
- The experimental initialization and settings could be better discussed and the details should be more clear. The evaluation metric of the results could be more comprehensive.
- Data Imbalance and Processing: The paper uses a downsampled dataset for one-year mortality prediction, which may introduce bias. It would benefit from a deeper exploration of how this impacts the model's performance.
- There is lack of ablation study to analyze the design.
- There is lack of analysis or discussion on how missing or incomplete modalities will affect the performance

**Questions:**

see weakness section.

---

### Official Review · Reviewer_nays · 2024-10-23

**Soundness:** 2
**Presentation:** 2
**Contribution:** 2
**Rating:** 3
**Confidence:** 4

**Summary:**

This paper presents a multimodal graph-based large language model (MG-LLM) designed for healthcare research. MG-LLM can simultaneously process four different modalities of electronic health records (EHRs) while incorporating graph learning techniques. The multimodal embeddings are then integrated into the LLM, leveraging its strengths in natural language processing and text reasoning. MG-LLM is evaluated on the MIMIC-IV and MIMIC-CXR datasets, demonstrating superior performance compared to baseline models. However, the paper has several significant drawbacks, and the evaluation results are not particularly convincing. For further details, please refer to the weaknesses and questions outlined in the review.

**Strengths:**

1. A promising solution for combining graph structures with LLMs in healthcare research.
2. A robust model for processing four modalities of EHRs.

**Weaknesses:**

1. The article's writing quality requires significant improvement, particularly in terms of consistency, completeness, and coherence of the logic.
2. Many of the references cited do not adequately support the authors' arguments, as the majority are not from peer-reviewed sources.
3. The paper lacks an introduction and analysis of state-of-the-art multimodal large language models (Multimodal-LLMs), such as BLIP, BLIP-2, and LLAVA.
4. As you have proposed a graph-related LLM, the Introduction section lacks any discussion or review of relevant graph-related research.
5. The method section lacks sufficient mathematical symbols and formulas to clearly illustrate your model, making it difficult to follow the model's pipeline.
6. In addition, the method section does not explain how this graph-LLM operates on similar graphs or temporal graphs.
7. There are several issues in the experimental results and analysis, which render the findings unconvincing.

**Questions:**

1. In the introduction, specifically in lines 41-46, you discuss the limitations of LLMs in processing multimodal data within the healthcare research field. However, several multimodal LLMs have already been developed for EHRs, such as Health-LLM [1] and Med-MLLM [2]. Additionally, models like BLIP2 [3] and LLAVA [4] can be adapted for healthcare research. Why do you not analyze your model's advancements in comparison to these existing models?
2. Why is there no discussion of any graph-related LLMs introduced in the first section, such as [5, 6]?
3. Why are there many unpublished research works included in your related-work section as references?
4. How can I gain a deeper understanding of the functions of temporal edges and similar edges used in your model? Could you provide an additional graph to illustrate this?
5. What does "residual stream" refer to in the method section, and what does the symbol "v" represent?
6. How is longitudinal patient EHR data utilized in your model? Is it truly effective for predicting patient mortality, given that the occurrence of death is fixed at the last time of the visit? There are some longutidal EHR analyzation works you could refer [7,8].
7. In the experiment section, why does the LLM Llama-3-8b exhibit the worst evaluation performance compared to all baseline models,
including non-LLM-based methods? Are there any potential mistakes in this evaluation in Table 1?
8. The highest F1 score in Table 1 is achieved by the model from Rezk et al. (2024). Why do you state that your model reached the highest F1 score in Section 4.2?

References:

[1].  Kim, Y., Xu, X., McDuff, D., Breazeal, C. &amp; Park, H.W.. (2024). Health-LLM: Large Language Models for Health Prediction via Wearable Sensor Data. <i>Proceedings of the fifth Conference on Health, Inference, and Learning</i>, in <i>Proceedings of Machine Learning Research</i> 248:522-539 Available from https://proceedings.mlr.press/v248/kim24b.html.

[2]. Liu, F., Zhu, T., Wu, X., Yang, B., You, C., Wang, C., ... & Clifton, D. A. (2023). A medical multimodal large language model for future pandemics. NPJ Digital Medicine, 6(1), 226.

[3]. Li, J., Li, D., Savarese, S., & Hoi, S. (2023, July). Blip-2: Bootstrapping language-image pre-training with frozen image encoders and large language models. In International conference on machine learning (pp. 19730-19742). PMLR.

[4]. Liu, H., Li, C., Wu, Q., & Lee, Y. J. (2024). Visual instruction tuning. Advances in neural information processing systems, 36.

[5]. Wang, H., Feng, S., He, T., Tan, Z., Han, X., & Tsvetkov, Y. (2024). Can language models solve graph problems in natural language?. Advances in Neural Information Processing Systems, 36.

[6]. Fatemi, B., Halcrow, J., & Perozzi, B. Talk like a Graph: Encoding Graphs for Large Language Models. In The Twelfth International Conference on Learning Representations.

[7]. Niu, S., Yin, Q., Ma, J., Song, Y., Xu, Y., Bai, L., ... & Yang, X. (2024). Enhancing healthcare decision support through explainable AI models for risk prediction. Decision Support Systems, 181, 114228.

[8]. Gao, J., Xiao, C., Wang, Y., Tang, W., Glass, L. M., & Sun, J. (2020, April). Stagenet: Stage-aware neural networks for health risk prediction. In Proceedings of The Web Conference 2020 (pp. 530-540).

**Details Of Ethics Concerns:**

The authors mention using the GPT-4 series model in their experiments as part of some baseline models with MIMIC data. However, PhysioNet has strict guidelines regarding the Responsible use of MIMIC data with online services such as GPT. The authors must clarify the source of the GPT model used and confirm that they have signed the "opt-outs for data sharing" form in compliance with these regulations.

---

### Official Review · Reviewer_hPW9 · 2024-10-30

**Soundness:** 2
**Presentation:** 2
**Contribution:** 2
**Rating:** 3
**Confidence:** 4

**Summary:**

This paper propose Multimodal Graph-LLM (MG-LLM) to utilize the multimodal information that is (1) across time for the same patient,  and (2) similar data from different patients. It combines multimodal data together with the Graph Neural Networks (GNNs) in order to enhance patient representation, improve multimodal data integration, and enable better contextual reasoning. This work in general is easy to understand and the idea presented is intuitive to follow. However, the work itself needs more to demonstrate its performance. The illustration also needs to be improved. I think it needs more work to meet the acceptance criteria.

**Strengths:**

1.	The general idea of the paper is easy to understand.

2.	The concept presented is intuitive and easy to follow.

**Weaknesses:**

1.	The experiments are not comprehensive and convincing. The only prediction task presented is a binary classification of one-year-mortality for the patient using the MIMIC-IV Data. More experiments would be needed to lead to a solid conclusion.

2.	Some justification and explanations are not clear. For example, how specifically is Eq.1 used for alignment is not clearly explained.

3.	Some experimental initialization and settings (e.g., embedding networks) could have been better established, and the details should be more clear (e.g., embedding dimensions, etc.). The evaluation metric of the results could be more comprehensive.

**Questions:**

1.	I do not understand how Eq. 1. Is used for the alignment process.  Is this loss function used to fine-tune the mapping network of the text and image embeddings and therefore update the embeddings then?

2.	It was mentioned that a ResNet50 model pre-trained on ImageNet was utilized to extract image features at the beginning. Considering the domain gap between the nature image and medical image, wouldn’t a medical-image focused network a better choice for this initialization?

3.	Why would the dominator of equation 1 distinguish between i = j and i != j？ They can be mathematically combined.

4.	What is dimension of the embedding vector spaces?

5.	What are lab results? Maybe an example/explanation could be provided for readers to better understand those data.

6.	For a classification task, it could be good to also include the AUC as an evaluation metric.

---

### Official Review · Reviewer_X42b · 2024-11-04

**Soundness:** 3
**Presentation:** 3
**Contribution:** 2
**Rating:** 5
**Confidence:** 4

**Summary:**

This paper presents an LLM-based framework, MG-LLM, that combines the large language model (LLM) with graph neural networks (GNNs) to make multimodal healthcare predictions. The framework constructs modality-specific graphs, uses GNNs for information propagation, performs multimodal alignment using contrastive learning, and injects the resulting context vectors into LLM. The paper evaluates the proposed approach on MIMIC-IV datasets for mortality prediction tasks, showing improved performance compared to baseline models.

**Strengths:**

1. The proposed model is sound.
2. The authors provide comparisons with multiple baseline models, including both LLM-based and non-LLM approaches.
3. The proposal achieves improved performance compared to baseline models.

**Weaknesses:**

1. The novelty of the framework needs clarification.
- Unifying multimodal graphs using contrastive learning has been widely explored (multiple literature can be found on Google Scholar).
- Using multiple types of graphs and multimodal data for healthcare has been studied in [1]. To my knowledge, using multimodal graphs to help LLMs better understand multimodal information appears to be novel.
- The idea of Patient Similarity Integration and Context Injection is similar to the retrieve and refine modules proposed in [3]. A discussion and comparison would be beneficial.
[1] GraphCare: Enhancing Healthcare Predictions with Personalized Knowledge Graphs. ICLR 2024.
[2] Multimodal machine learning in precision health: a scoping review. npj Digital Medicine, 2022.
[3] Retrieve, Reason, and Refine: Generating Accurate and Faithful Patient Instructions. NeurIPS, 2022.
2. Several important implementation and experimental details are unclear:
- The statistics of the pre-processed dataset are not provided. Since the evaluation dataset is self-processed rather than from the original dataset, it is important to provide details of the processed dataset. What is the total number of data used for training and evaluation? Meanwhile, how are the results of previous methods obtained on the processed dataset?
- The graph details need clarification: Is it a directed or undirected graph? How to calculate edge weights? How to construct a node - is it the embedding of the entire data for a patient?
- The architecture of the GNNs used is not specified.
- The contrastive learning details are unclear, including the value of τ and the number and ratio of positive and negative samples.
- The paper lacks many implementation details necessary for reproducibility: Important hyperparameters such as learning rate, epochs, and batch size are missing. Since the paper uses LLMs as the backbone, details about LoRA usage for model training and its corresponding hyperparameters should be provided. Additionally, details about training time, inference time, and GPU hours required are important.

3. Several important insights are missing:
- Why can the model generate better explanations when it is trained for binary predictions?
- The evaluation dataset has a positive example rate of 0.23, suggesting that an algorithm predicting all negative samples would achieve 77% accuracy. Many models in Table 1 perform below 77%. More insights on this observation are needed.
- How does the model handle missing modalities/data during inference?

4. The analysis requires improvement:
- The ablation studies focus only on modality combinations, lacking analysis of different graph construction methods, injection points in LLM layers, and the effect of different GNN architectures on performance.
- Only one prediction task (mortality) is evaluated. The statistical significance of performance improvements needs discussion, especially given the imbalanced evaluation data.
- The significant biases in the dataset are not discussed.

5. Several additional experiments are strongly recommended:
- Statistical significance tests.
- Systematic error analysis.

**Questions:**

Copied from Weaknesses:

1. The novelty of the framework needs clarification.
- Unifying multimodal graphs using contrastive learning has been widely explored (multiple literature can be found on Google Scholar).
- Using multiple types of graphs and multimodal data for healthcare has been studied in [1]. To my knowledge, using multimodal graphs to help LLMs better understand multimodal information appears to be novel.
- The idea of Patient Similarity Integration and Context Injection is similar to the retrieve and refine modules proposed in [3]. A discussion and comparison would be beneficial.
[1] GraphCare: Enhancing Healthcare Predictions with Personalized Knowledge Graphs. ICLR 2024.
[2] Multimodal machine learning in precision health: a scoping review. npj Digital Medicine, 2022.
[3] Retrieve, Reason, and Refine: Generating Accurate and Faithful Patient Instructions. NeurIPS, 2022.
2. Several important implementation and experimental details are unclear:
- The statistics of the pre-processed dataset are not provided. Since the evaluation dataset is self-processed rather than from the original dataset, it is important to provide details of the processed dataset. What is the total number of data used for training and evaluation? Meanwhile, how are the results of previous methods obtained on the processed dataset?
- The graph details need clarification: Is it a directed or undirected graph? How to calculate edge weights? How to construct a node - is it the embedding of the entire data for a patient?
- The architecture of the GNNs used is not specified.
- The contrastive learning details are unclear, including the value of τ and the number and ratio of positive and negative samples.
- The paper lacks many implementation details necessary for reproducibility: Important hyperparameters such as learning rate, epochs, and batch size are missing. Since the paper uses LLMs as the backbone, details about LoRA usage for model training and its corresponding hyperparameters should be provided. Additionally, details about training time, inference time, and GPU hours required are important.

3. Several important insights are missing:
- Why can the model generate better explanations when it is trained for binary predictions?
- The evaluation dataset has a positive example rate of 0.23, suggesting that an algorithm predicting all negative samples would achieve 77% accuracy. Many models in Table 1 perform below 77%. More insights on this observation are needed.
- How does the model handle missing modalities/data during inference?

4. The analysis requires improvement:
- The ablation studies focus only on modality combinations, lacking analysis of different graph construction methods, injection points in LLM layers, and the effect of different GNN architectures on performance.
- Only one prediction task (mortality) is evaluated. The statistical significance of performance improvements needs discussion, especially given the imbalanced evaluation data.
- The significant biases in the dataset are not discussed.

5. Several additional experiments are strongly recommended:
- Statistical significance tests.
- Systematic error analysis.

---

### Note · Authors · 2024-11-27

**Comment:**

We are withdrawing this submission. We appreciate the time and effort of the reviewers.

**Withdrawal Confirmation:**

I have read and agree with the venue's withdrawal policy on behalf of myself and my co-authors.